# On a 2-Relative Entropy

**DOI:** 10.3390/e24010074

**Published:** 2021-12-31

**Authors:** James Fullwood

**Affiliations:** School of Mathematical Sciences, Shanghai Jiao Tong University, 800 Dongchuan Road, Shanghai 200240, China; fullwood@sjtu.edu.cn

**Keywords:** 2-category, Bayesian inference, discrete memoryless channel, functor, information theory, relative entropy, synthetic probability, primary 94A17, secondary 18A05, 62F15

## Abstract

We construct a 2-categorical extension of the relative entropy functor of Baez and Fritz, and show that our construction is functorial with respect to vertical morphisms. Moreover, we show such a ‘2-relative entropy’ satisfies natural 2-categorial analogues of convex linearity, vanishing under optimal hypotheses, and lower semicontinuity. While relative entropy is a relative measure of information between probability distributions, we view our construction as a relative measure of information between *channels*.

## 1. Introduction

Let *X* and *Y* be finite sets which are the input and output alphabets of a discrete memoryless channel $X\overset{f}{⇝}Y$ with probability transition matrix $f_{yx}$, representing the probability of the output *y* given the input *x*. Every input *x* then determines a probability distribution on *Y* which we denote by $f^{x}$, so that $f^{x}\left(y\right)=f_{yx}$ for all x∈X and y∈Y. The channel $X\overset{f}{⇝}Y$ together with the choice of a prior distribution *p* on *X* will be denoted (f|p), and such data then determine a distribution ϑ(f|p) on X×Y given by $\vartheta {\left(f|p\right)}_{\left(x,y\right)}=p_{x}f_{yx}$. Given a second channel $X\overset{g}{⇝}Y$ with prior distribution *q* on *X*, the chain rule for relative entropy says that the relative entropy $D\left(\vartheta (f|p),\vartheta (g|q)\right)$ is given by
(1)$$D\left(\vartheta (f|p),\vartheta (g|q)\right)=D\left(p,q\right)+\sum_{x\in X}p_{x}D\left(f^{x},g^{x}\right).$$
As the RHS of (Equation 1) involves precisely the datum of the channels *f* and *g* together with the prior distributions *p* and *q*, we view the quantity $D\left(\vartheta (f|p),\vartheta (g|q)\right)$ as a relative measure of information between the *channels*(f|p) and (g|q). In particular, since from a Bayesian perspective D(p,q) may be thought of as the amount of information gained upon discovering that the assumed prior distribution *p* is actually *q*, it seems only natural to think of $D\left(\vartheta (f|p),\vartheta (g|q)\right)$ as the amount of information gained upon learning that the assumed channel (f|p) is actually the channel (g|q).

To make such a Bayesian interpretation more precise, we build upon the work of Baez and Fritz [1], who formulated a type of Bayesian inference as a *process* X→Y (including a set of conditional hypotheses on the outcome of the process), which given a prior distribution *p* on *X* yields distributions *r* on *Y* and *q* on *X* in such a way that the relative entropy D(p,q) has an operational meaning as a quantity associated with a Bayesian updating with respect to the process X→Y. (Here, *X* may be thought of more generally as the set of possible states of some system to be measured, while *Y* may be thought of as the possible outcomes of the measurement.) Baez and Fritz then proved that up to a constant multiple, the map on such processes given by
(2)(X→Y)↦RE(X→Y):=D(p,q)
is the unique map satisfying the following axioms.

**Functoriality**: Given a composition of processes X→Y→Z,
RE(X→Y→Z)=RE(X→Y)+RE(Y→Z).**Convex Linearity**: Given a collection of processes $U^{x}\to V^{x}$ indexed by the elements x∈X of a finite probability space (X,p),
$$\mathrm{RE}\left(\sum_{x\in X}p_{x}\left(U^{x}\to V^{x}\right)\right)=\sum_{x\in X}p_{x}\mathrm{RE}\left(U^{x}\to V^{x}\right).$$**Vanishing Under Optimal Hypotheses**: If the conditional hypotheses associated with a process X→Y are optimal, then
RE(X→Y)=0.**Continuity**: The map (X→Y)↦RE(X→Y) is lower semi-continuous.

While Baez and Fritz facilitated their exposition using the language of category theory [2], knowing that a category consists of a class of objects together with a class of composable arrows (i.e., morphisms) between objects is all that is needed for an appreciation of their construction. From such a perspective, the aforementioned processes X→Y are morphisms in a category FinStat, and the relative entropy assignment given by (Equation 2) is then a map from morphisms in FinStat to [0,∞].

In what follows, we elevate the construction of Baez and Fritz to the level of 2-categories (or more precisely, *double categories*), whose 2-morphisms may be viewed as certain processes between processes, or rather, processes which connect one channel to another. In particular, in Section 2, we formally introduce the category FinStat introduced by Baez and Fritz, and then, in Section 3, we review their functorial characterization of relative entropy using FinStat. In Section 4, we construct a category ${\mathrm{FinStat}}_{2}$ which is a 2-level extension of FinStat, and in Section 5, we define a convex structure on ${\mathrm{FinStat}}_{2}$. In Section 6, we define a relative measure of information between channels which we refer to as *conditional relative entropy*, and show that it is convex linear and functorial with respect to vertical morphisms in ${\mathrm{FinStat}}_{2}$. The conditional relative entropy is then used in Section 7 to define a relative entropy assignment ${\mathrm{RE}}_{2}$ on 2-morphisms via the chain rule as given by (Equation 1) (for more on the chain rule for relative entropy one may consult Chapter 2 of [3]). Moreover, we show that such a ‘2-relative entropy’ satisfies the natural 2-level analogues of axioms 1–4 as satisfied by the relative entropy map RE.

As abstract as a relative entropy of processes between processes may seem, Shannon’s Noisy Channel Coding Theorem—which is a cornerstone of information theory—is essentially a statement about transforming a noisy channel into a noiseless one via a sequence of codings and encodings. From such a viewpoint, information theory is fundamentally about processes (i.e., a sequence of codings and encodings), between processes (i.e., channels), and it is precisely this viewpoint with which we will proceed. Furthermore, there is a growing recent interest in axiomatic and categorical approaches to information theory [4,5,6,7,8,9,10,11,12,13], and the present work is a direct outgrowth of such activity.

## 2. The Category FinStat

In this section, we introduce the first-level structure of interest, which is the category FinStat introduced by Baez and Fritz [1]. Though we use the language of categories, knowing that a category consists of a class of composable arrows between a class of objects is sufficient for the comprehension of all categorical notions in this work.

**Definition** **1.**
*Let X and Y be finite sets. A*
**discrete memoryless channel**
*(or simply*
**channel**
*for short) $X\overset{f}{⇝}Y$ associates every x∈X with a probability distribution $f^{x}$ on Y. In such a case, the sets X and Y are referred to as the*
**set of inputs**
*and*
**set of outputs**
*of the channel f, respectively, and $f^{x}\left(y\right)$ is the probability of receiving the output y given the input x, which will be denoted by $f_{yx}$.*


**Definition** **2.**
*If $X\overset{f}{⇝}Y$ and $Y\overset{g}{⇝}Z$ are channels, then the composition $X\overset{g\circ f}{⇝}Z$ is given by*

$$g_{zx}=\sum_{x\in X}g_{zy}f_{yx}$$

*for all x∈X and y∈Y.*


**Remark** **1.**
*If $X\overset{f}{⇝}Y$ is a channel such that for every x∈X there exists a y∈Y with $f_{yx}=1$, then such a y is necessarily unique given x, and as such, f may be identified with a function f:X→Y. In such a case, we say that the channel f is*
**pure**
*(or*
**deterministic**
*), and from here on, we will not distinguish the difference between a pure channel and the associated function from its set of inputs to its set of outputs.*


**Definition** **3.***If* ★ *denotes a set with a single element, then a channel $★\overset{p}{⇝}X$ is simply a probability distribution on X, and, in such a case, we will use $p_{x}$ to denote the probability of x as given by p for all x∈X. The pair (X,p) is then referred to as a*
**finite probability space**.


**Notation** **1.**
*The datum of a channel $X\overset{f}{⇝}Y$ together with a prior distribution $★\overset{p}{⇝}X$ on its set of inputs will be denoted (f|p).*


**Definition** **4.**
*Let FinStat denote the category whose objects are finite probability spaces, and whose morphisms (X,p)⟶(Y,q) consist of the following data:*

*A function f:X→Y such that f∘p=q;*

*A channel $Y\overset{s}{⇝}X$ such that $f\circ s={\mathrm{id}}_{Y}$. In other words, $Y\overset{s}{⇝}X$ is a*
**stochastic section**
*of f:X→Y.*

*A morphism in FinStat is then summarized by a diagram of the form*

(3)

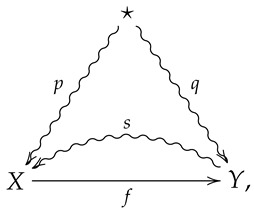


*and a composition of morphisms in FinStat is obtained via function composition and composition of stochastic sections. In such a case, it is straightforward to show that a composition of stochastic sections is a stochastic section, etc. The morphism corresponding to diagram (Equation 3) will often be denoted (f,p,s).*


**Remark** **2.**
*Note that in diagram (Equation 3), a straight arrow is used for $X\overset{f}{⟶}Y$ as f is a function, as opposed to a noisy channel.*


**Remark** **3.**
*The operational interpretation of diagram (Equation 3) is as follows. The set X is thought of as the set of possible states of the system, and f:X→Y is then thought of as a measurement process, so that Y is then thought of as the set of possible states of some measuring apparatus. The stochastic section $Y\overset{s}{⇝}X$ is then thought of as a set of hypotheses about the state of the system given a state of the measuring apparatus. In particular, $s_{xy}$ is thought of as the probability the system was in state x given the state y of the measuring apparatus.*


**Definition** **5.**
*If the stochastic section $Y\overset{s}{⇝}X$ in diagram (Equation 3) is such that s∘q=p, then s will be referred to as an*
**optimal hypothesis**
*for (f|p).*


**Definition** **6.**
*Let (X,p) be a finite probability space, and let $U^{x}\overset{\mu^{x}}{⇝}V^{x}$ be a collection of channels with prior distributions $★\overset{q^{x}}{⇝}U^{x}$ indexed by X. The*
**convex combination**
*of $\left(\mu^{x}|q^{x}\right)$ with respect to (X,p) is the channel*

$$\left.\left(\underset{x\in X}{∐}U^{x}\overset{\;{⨁}_{x\in X}\mu^{x}\;}{⇝}\underset{x\in X}{∐}V^{x}\right|\underset{x\in X}{⨁}p_{x}q^{x}\right),$$

*where ${⨁}_{x\in X}\mu^{x}$ is the channel given by*

$${\left(\underset{x\in X}{⨁}\mu^{x}\right)}_{vu}=\left\{\begin{array}{c} \mu_{vu}^{x}\;\mathrm{if}\;\left(v,u\right)\in V^{x}\times U^{x}\;\mathrm{for}\;\mathrm{some}\;x\in X\\ 0\;\;\mathrm{otherwise} \end{array}\right.,$$

*with prior distribution $★\overset{\;{⨁}_{x\in X}p_{x}q^{x}\;}{⇝}∐U^{x}$ given by*

$${\left(\underset{x\in X}{⨁}p_{x}q^{x}\right)}_{u}=p_{x_{u}}q_{u}^{x_{u}},$$

*where $x_{u}$ is such that $u\in U^{x_{u}}$. Such a convex combination will be denoted ${⨁}_{x\in X}p_{x}\left(\mu^{x}|q^{x}\right)$.*


## 3. The Baez and Fritz Characterization of Relative Entropy

We now recall the Baez and Fritz characterization of relative entropy in FinStat.

**Definition** **7.**
*Let (X,p) be a finite probability space, and let*

$$\left(U^{x}\overset{\mu^{x}}{⟶}V^{x},★\overset{q^{x}}{⇝}U^{x},V^{x}\overset{s^{x}}{⇝}U^{x}\right)$$

*be a collection of morphisms in FinStat indexed by X. The*
**convex combination**
*of $\left(\mu^{x},q^{x},s^{x}\right)$ with respect to (X,p) is the morphism ${⨁}_{x\in X}p_{x}\left(\mu^{x},q^{x},s^{x}\right)$ in FinStat corresponding to the diagram*

(4)

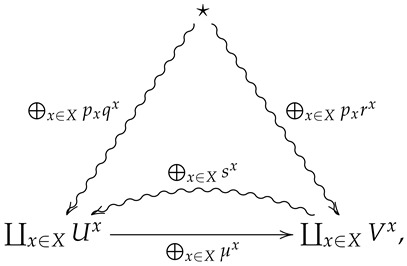


*where $r^{x}=\mu^{x}\circ q^{x}$ for all x∈X.*


**Definition** **8.**
*Let (f,p,s) be a morphism in FinStat, and let r=s∘f∘p. The*
**relative entropy**
*of (f,p,s) is the non-negative extended real number RE(f,p,s)∈[0,∞] given by*

RE(f,p,s)=D(p,r),

*where $D\left(p,r\right)=\sum_{x}p_{x}\log\left(p_{x}/r_{x}\right)$ is the relative entropy between the distributions p and r on X.*


**Definition** **9.**
*Let F:FinStat→[0,∞] be a map from the morphisms in FinStat to the extended non-negative reals [0,∞].*

*F is said to be*
**functorial**
*if and only if for every composition (g∘f,p,s∘t) of morphisms in FinStat we have*

F(g∘f,p,s∘t)=F(f,p,s)+F(g,f∘p,t).


*F is said to be*
**convex linear**
*if and only if for every convex combination ${⨁}_{x\in X}$$p_{x}\left(\mu^{x},q^{x},s^{x}\right)$ of morphisms in FinStat we have*

$$F\left(\underset{x\in X}{⨁}p_{x}\left(\mu^{x},q^{x},s^{x}\right)\right)=\sum_{x\in X}p_{x}F\left(\mu^{x},q^{x},s^{x}\right).$$


*F is said to be*
**vanishing under optimal hypotheses**
*if and only if for every morphism (f,p,s) in FinStat with s an optimal hypothesis we have*

F(f,p,s)=0.


*F is said to be*
**lower semicontinuous**
*if and only if for every sequence of morphisms $\left(f,p_{n},s_{n}\right)$ in FinStat converging to a morphism (f,p,s) we have*

$$F\left(f,p,s\right)\leq \underset{n\to \infty }{lim inf}F\left(f,p_{n},s_{n}\right).$$




**Theorem** **1**(The Baez and Fritz Characterization of Relative Entropy)**.**
*Let S be the collection of maps from the morphisms in FinStat to [0,∞] which are functorial, convex linear, vanishing under optimal hypotheses and lower semicontinuous. Then, the following statements hold.*
*1*.*The relative entropy RE is an element of S;**2*.*If F∈S, then F=cRE for some non-negative constant c∈R.*

## 4. The Category ${\mathrm{FinStat}}_{2}$

In this section, we introduce the second-level structure of interest, namely, the double category ${\mathrm{FinStat}}_{2}$, which is a 2-level extension of FinStat.

**Definition** **10.**
*Let ${\mathrm{FinStat}}_{2}$ denote the 2-category whose objects and 1-morphisms coincide with those of FinStat, and whose 2-morphisms are constructed as follows. Given 1-morphisms*


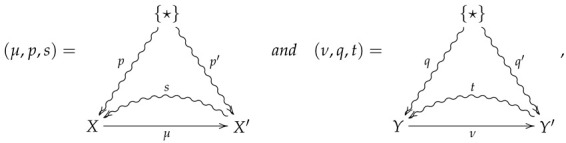


*a 2-morphism ♠:(μ,p,s)⇒(ν,q,t) consists of channels $X\overset{f}{⇝}Y$ and $X^{\prime }\overset{f^{\prime }}{⇝}Y^{\prime }$ such that*


f∘p=q



$f^{\prime }\circ p^{\prime }=q^{\prime }$



$\nu \circ f=f^{\prime }\circ \mu$


*The 2-morphism ♠:(μ,p,s)⇒(ν,q,t) may then be summarized by the following diagram.*

(5)

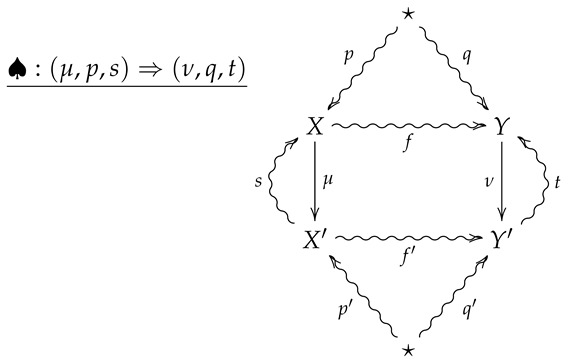




**Remark** **4.**
*A crucial point is that in the above diagram, all arrows necessarily commute except for any compositions involving the the outer ‘wings’, s and t. For example, the compositions s∘μ∘p and $t\circ f^{\prime }\circ \mu$ need not be equal to p and f, respectively.*


**Remark** **5.**
*Diagram (Equation 5) should be thought of as a flattened out pyramid, whose base is the inner square and whose vertex is obtained by the identification of the upper and lower stars in the diagram.*


**Remark** **6.**
*For an operational interpretation of a 2-morphism in ${\mathrm{FinStat}}_{2}$ as given by diagram (Equation 5), one may consider X and Y as sample spaces associated with all possible outcomes of experiments $E_{X}$ and $E_{Y}$. As the sets X and Y are endowed with prior distributions p and q, the maps $\mu :X\to X^{\prime }$ and $\nu :Y\to Y^{\prime }$ are then random variables with values in $X^{\prime }$ and $Y^{\prime }$, and the stochastic sections s and t then represent conditional hypotheses about the outcomes of the measurements corresponding to μ and ν. The channels f:X⇝Y and $f:X^{\prime }⇝Y^{\prime }$ then represent stochastic processes such that taking the measurement μ followed by $f^{\prime }$ results in the same process as first letting X evolve according to f and then taking the measurement ν.*


**Example** **1.**
*For a real-life scenario which realizes a 2-morphism in FinStat, suppose two experimenters Alice and Bob are collaborating on a project to verify predictions of a theory. As such, Alice and her data analyst partner Alicia travel to a mountain in Brazil during a solar eclipse to perform experiments while Bob and and his data-analyst partner Bernie travel to a mountain in Montenegro at the same time for the same purpose. Alice and Bob will then perform experiments in their separate locations and hand their results over to Alicia and Bernie, who will then analyze the data to produce numerical results. At the end of each day, Alice will report her results to Bob over a noisy channel while Alicia will report her results to Bernie over a noisy channel, so that Bob and Bernie may compare their results with Alice and Alicia’s. We then summarize such a scenario with the following 2-morphism in ${\mathrm{FinStat}}_{2}$:*

(6)

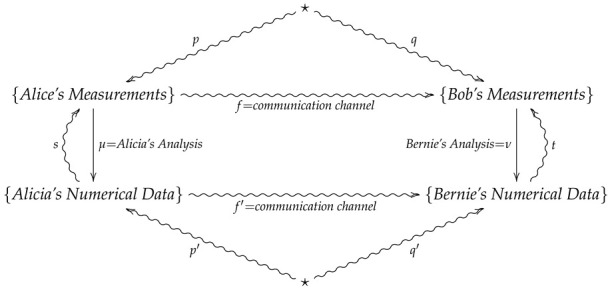


In diagram (Equation 6), *p* and *q* are assumed prior distributions on Alice and Bob’s measurements, while *s* and *t* are empirical conditional distributions on Alice and Bob’s measurements given the data outcomes of Alicia and Bernie’s analysis. Moreover, if the communication channel *f* is less reliable than $f^{\prime }$, then the composition $t\circ f^{\prime }\circ \mu$ provides a Bayesian updating for the channel *f*.

For vertical composition of 2-morphisms, suppose $♣:\left(\mu^{\prime },p^{\prime },s^{\prime }\right)\Rightarrow \left(\nu^{\prime },q^{\prime },t^{\prime }\right)$ is the 2-morphism summarized by the following diagram.

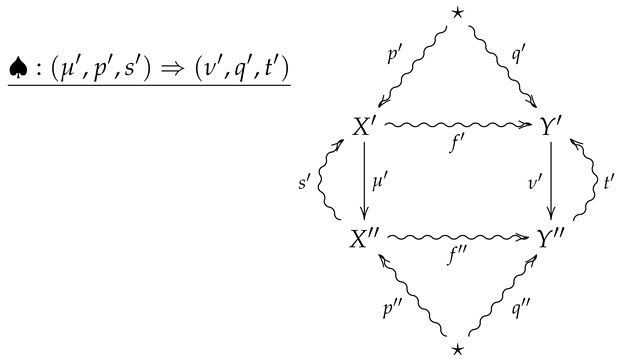


The vertical composition ♣∘♠ is then summarized by the following diagram.

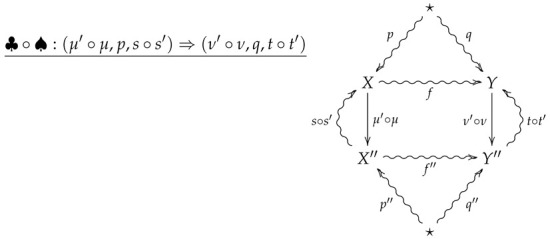


For horizontal composition, let ♡:(ν,q,t)⇒(ξ,r,u) be a 2-morphism summarized by the following diagram.

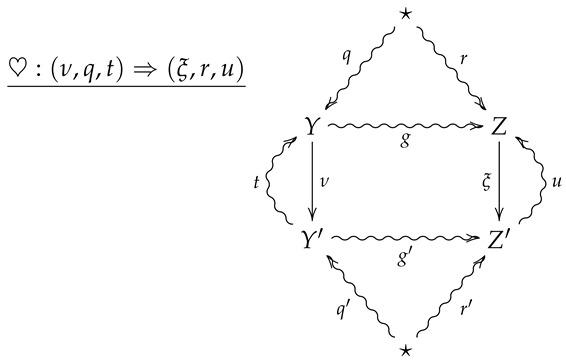


The horizontal composition ♡∘♠ is then summarized by the following diagram.

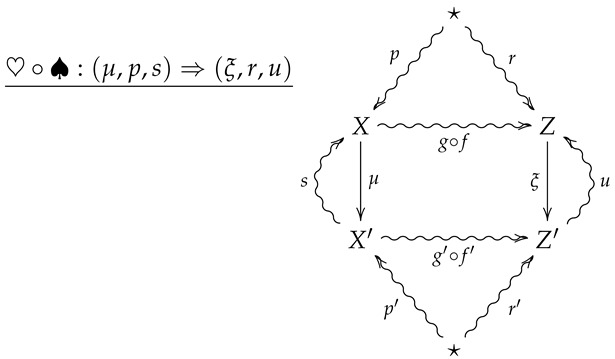


## 5. Convexity in ${\mathrm{FinStat}}_{2}$

We now generalize the convex structure on morphisms in FinStat to 2-morphisms in ${\mathrm{FinStat}}_{2}$. For this, let (X,p) be a finite probability space, and let ${♠}^{x}$ be a collection of 2-morphisms in ${\mathrm{FinStat}}_{2}$ indexed by *X*, where ${♠}^{x}$ is summarized by the following diagram.

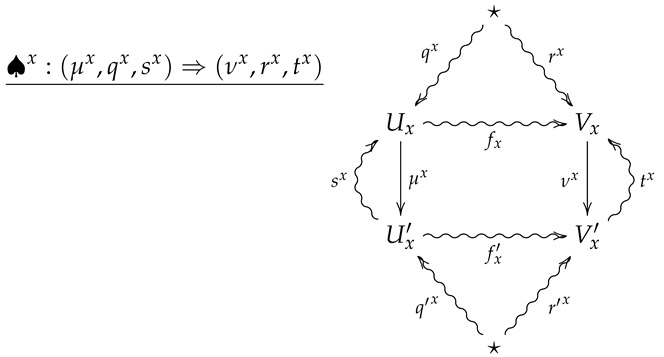


**Definition** **11.**
*The*
**convex sum**
*${⨁}_{x\in X}p_{x}{♠}^{x}$ is the 2-morphism in ${\mathrm{FinStat}}_{2}$ summarized by the following diagram.*


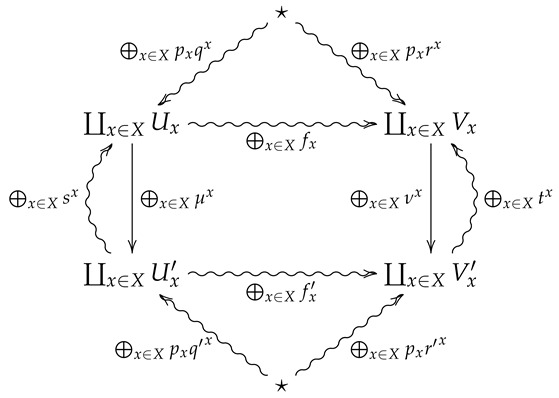




## 6. Conditional Relative Entropy in ${\mathrm{FinStat}}_{2}$

We now introduce a measure of information associated with 2-morphisms in ${\mathrm{FinStat}}_{2}$ which we refer to as ‘conditional relative entropy’. The results proved in this section are essentially all lemmas for the results proved in the next section, where we introduce a 2-level extension of the relative entropy map RE, and show that it satisfies the 2-level analogues of the characterizing axioms of relative entropy.

**Definition** **12.***With every 2-morphism*
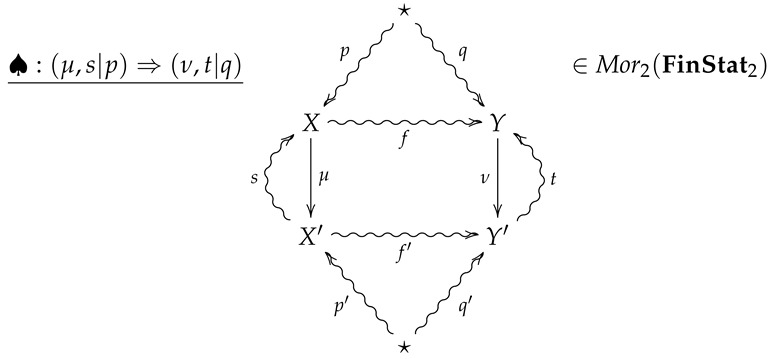

we associate the non-negative extended real number CE(♠)∈[0,∞] given by
(7)$$\mathrm{CE}\left(♠\right)=\sum_{x\in X}p_{x}D\left(f^{x},{\left(t\circ f^{\prime }\circ \mu \right)}^{x}\right),$$
where D(−,−) is the standard relative entropy. We refer to CE(♠) as the ***conditional relative entropy*** of ♠.

**Remark** **7.***We refer to CE(♠) as conditional relative entropy as its defining formula (Equation 7) is structurally similar to the defining formula for conditional entropy. In particular, if $X\overset{f}{⇝}Y$ is a channel with prior distribution $★\overset{p}{⇝}X$, then the conditional entropy H(f|p) is given by*$$H\left(f|p\right)=\sum_{x\in X}p_{x}H\left(f^{x}\right),$$
where $H(f^{x})$ is the Shannon entropy of the distribution $f^{x}$ on *Y*.

**Proposition** **1.**
*Conditional relative entropy in ${\mathrm{FinStat}}_{2}$ is convex linear, i.e., if (X,p) is a finite probability space and ${♠}^{x}$ is a collection of 2-morphisms in ${\mathrm{FinStat}}_{2}$ indexed by X, then*

$$CE\left(\underset{x\in X}{⨁}p_{x}{♠}^{x}\right)=\sum_{x\in X}p_{x}CE\left({♠}^{x}\right).$$



**Proof.** Suppose ⨁x∈Xpx♠x is summarized by the following diagram

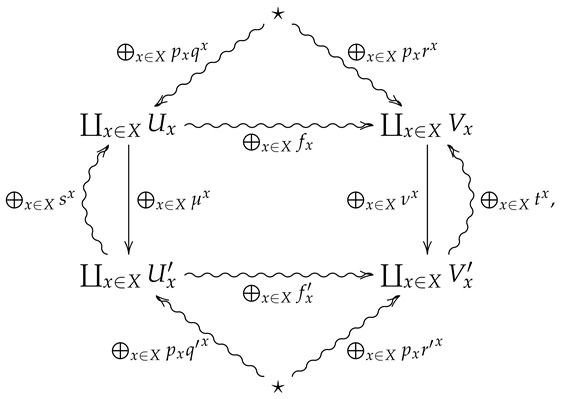

and let $U={∐}_{x\in X}U_{x}$, $\mu ={⊕}_{x\in X}\mu^{x}$, $f={⊕}_{x\in X}f^{x}$, $f^{\prime }={⊕}_{x\in X}f^{\prime x}$ and $t={⊕}_{x\in X}t^{x}$. We then have
$$\begin{array}{ccc} \mathrm{CE}\left(\underset{x\in X}{⨁}p_{x}{♠}^{x}\right) & = & \sum_{u\in U}q_{u}D\left(f^{u},{\left(t\circ f^{\prime }\circ \mu \right)}^{u}\right)\\ & = & \sum_{x\in X}\sum_{u_{x}\in U_{x}}p_{x}q_{u_{x}}^{x}D\left(f^{u_{x}},{\left(t\circ f^{\prime }\circ \mu \right)}^{u_{x}}\right)\\ & = & \sum_{x\in X}p_{x}\sum_{u_{x}\in U_{x}}q_{u_{x}}^{x}D\left({\left(f_{x}\right)}^{u_{x}},{\left(t^{x}\circ f_{x}^{\prime }\circ \mu^{x}\right)}^{u_{x}}\right)\\ & = & \sum_{x\in X}p_{x}\mathrm{CE}\left({♠}^{x}\right), \end{array}$$
as desired. □

**Theorem** **2.**
*Conditional relative entropy in ${\mathrm{FinStat}}_{2}$ is functorial with respect to vertical composition, i.e., if ♣∘♠ is a vertical composition in ${\mathrm{FinStat}}_{2}$, then CE(♣∘♠)=CE(♣)+CE(♠).*


**Lemma** **1.**
*Let $X\overset{f}{⇝}Y\overset{g}{⇝}Z$ be a composition of channels.*
*1*.
*If f is a pure channel, then ${\left(g\circ f\right)}_{zx}=g_{zf(x)}$;*
*2*.
*If g is a stochastic section of a pure channel $Z\overset{h}{\to }Y$, then ${\left(g\circ f\right)}_{zx}=g_{zh(z)}f_{h(z)x}$.*



**Proof.** The statements 1 and 2 follow immediately from the definitions of pure channel and stochastic section. □

**Lemma** **2.**
*Let ♠ be a 2-morphism in ${\mathrm{FinStat}}_{2}$ as summarized by the diagram*


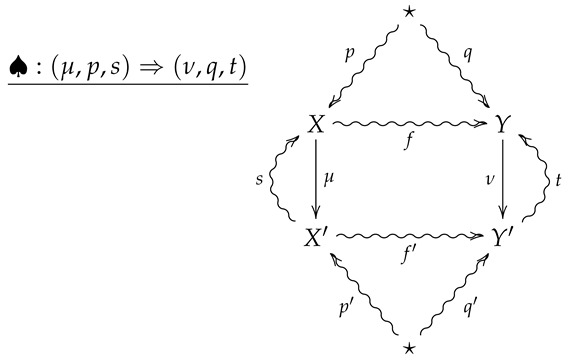


*Then,*
*1*.
*$CE\left(♠\right)=\sum_{x\in X}\sum_{y\in Y}p_{x}f_{yx}\log\left(\frac{f_{yx}}{t_{y\nu (y)}f_{\nu (y)\mu (x)}^{\prime }}\right)$ for all x∈X;*
*2*.
*$p_{x^{\prime }}^{\prime }f_{y^{\prime }x^{\prime }}^{\prime }=\sum_{x\in \mu^{-1}\left(x^{\prime }\right)}\sum_{y\in \nu^{-1}\left(y^{\prime }\right)}p_{x}f_{yx}$ for all $\left(x^{\prime },y^{\prime }\right)\in X^{\prime }\times Y^{\prime }$.*



**Proof.** To prove item (1), let x∈X and y∈Y. Then,
$$\begin{array}{ccc} {\left(t\circ f^{\prime }\circ \mu \right)}_{yx} & = & {\left(t\circ \left(f^{\prime }\circ \mu \right)\right)}_{yx}\\ & = & \sum_{y^{\prime }\in Y^{\prime }}t_{yy^{\prime }}{\left(f^{\prime }\circ \mu \right)}_{y^{\prime }x}\\ & = & \sum_{y^{\prime }\in Y}t_{yy^{\prime }}f_{y^{\prime }\mu \left(x\right)}^{\prime }\\ & = & t_{y\nu (y)}f_{\nu (y)\mu (x)}^{\prime }, \end{array}$$
where the third equality follows from Lemma 1 since μ is a pure channel, and the fourth equality follows also from Lemma 1 since *t* is a stochastic section of a pure channel. We then have
$$\begin{array}{ccc} \mathrm{CE}(♠) & = & \sum_{x\in X}p_{x}D\left(f^{x},{\left(t\circ f^{\prime }\circ \mu \right)}^{x}\right)\\ & = & \sum_{x\in X}\sum_{y\in Y}p_{x}f_{yx}\log\left(\frac{f_{yx}}{{\left(t\circ f^{\prime }\circ \mu \right)}_{yx}}\right)\\ & = & \sum_{x\in X}\sum_{y\in Y}p_{x}f_{yx}\log\left(\frac{f_{yx}}{t_{y\nu (y)}f_{\nu (y)\mu (x)}^{\prime }}\right), \end{array}$$
as desired.To prove item (2), the condition $\nu \circ f=f^{\prime }\circ \mu$ is equivalent to the equation ${\left(\nu \circ f\right)}_{y^{\prime }x}={\left(f^{\prime }\circ \mu \right)}_{y^{\prime }x}$ for all $y^{\prime }\in Y^{\prime }$ and x∈X. In addition, since
$${\left(f^{\prime }\circ \mu \right)}_{y^{\prime }x}=f_{y^{\prime }\mu \left(x\right)}^{\prime },$$
and
$${\left(\nu \circ f\right)}_{y^{\prime }x}=\sum_{y\in Y}\nu_{y^{\prime }y}f_{yx}=\sum_{y\in \nu^{-1}\left(y^{\prime }\right)}\nu_{y^{\prime }y}f_{yx}=\sum_{y\in \nu^{-1}\left(y^{\prime }\right)}f_{yx},$$
it follows that $f_{y^{\prime }\mu \left(x\right)}^{\prime }=\sum_{y\in \nu^{-1}\left(y^{\prime }\right)}f_{yx}$, thus, for all $x^{\prime }\in X^{\prime }$ and $y^{\prime }\in Y$, it follows that $f_{y^{\prime }x^{\prime }}^{\prime }=\sum_{y\in \nu^{-1}\left(y^{\prime }\right)}f_{yx}$ for all $x\in \mu^{-1}\left(x^{\prime }\right)$. As such, we have
$$p_{x^{\prime }}^{\prime }f_{y^{\prime }x^{\prime }}^{\prime }=\left(\sum_{x\in \mu^{-1}\left(x^{\prime }\right)}p_{x}\right)\left(\sum_{y\in \nu^{-1}\left(y^{\prime }\right)}f_{yx}\right)=\sum_{x\in \mu^{-1}\left(x^{\prime }\right)}\sum_{y\in \nu^{-1}\left(y^{\prime }\right)}p_{x}f_{yx},$$
as desired. □

**Proof** **of Theorem 2.**Suppose ♠ and ♣ are such that the vertical composition ♣∘♠ is summarized by the following diagram.

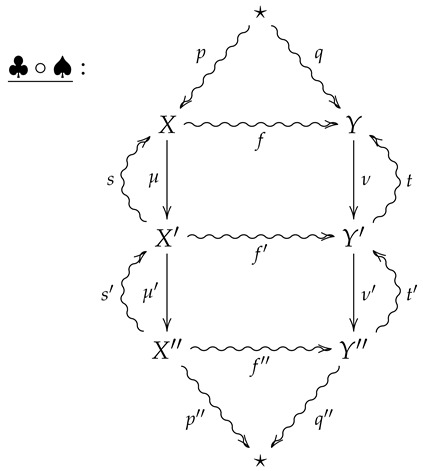

By item (1) in Lemma 2, we then have
$$\mathrm{CE}\left(♣\circ ♠\right)=\sum_{x\in X}\sum_{y\in Y}p_{x}f_{yx}\log\left(\frac{f_{yx}}{{\left(t\circ t^{\prime }\right)}_{y\left(\nu^{\prime }\circ \nu \right)\left(y\right)}f_{\left(\nu^{\prime }\circ \nu \right)\left(y\right)\left(\mu^{\prime }\circ \mu \right)\left(x\right)}^{\prime \prime }}\right),$$
and since *t* is a section of the pure channel ν, by item (2) of Lemma 1, it follows that for every y∈Y we have
$${\left(t\circ t^{\prime }\right)}_{y\left(\nu^{\prime }\circ \nu \right)\left(y\right)}=t_{y\nu (y)}t_{\nu \left(y\right)\left(\nu^{\prime }\circ \nu \right)\left(y\right)}^{\prime }.$$As such, we have
$$\begin{array}{ccc} \mathrm{CE}(♣\circ ♠) & = & \sum_{x\in X}\sum_{y\in Y}p_{x}f_{yx}\log\left(\frac{f_{yx}}{t_{y\nu (y)}t_{\nu \left(y\right)\left(\nu^{\prime }\circ \nu \right)\left(y\right)}^{\prime }f_{\left(\nu^{\prime }\circ \nu \right)\left(y\right)}^{\prime \prime }}\right)\\ & = & \sum_{x\in X}\sum_{y\in Y}p_{x}f_{yx}\log\left(\frac{f_{yx}f_{\nu (y)\mu (x)}^{\prime }}{t_{y\nu (y)}f_{\nu (y)\mu (x)}^{\prime }t_{\nu \left(y\right)\left(\nu^{\prime }\circ \nu \right)\left(y\right)}^{\prime }f_{\left(\nu^{\prime }\circ \nu \right)\left(y\right)\left(\mu^{\prime }\circ \mu \right)\left(x\right)}^{\prime \prime }}\right)\\ & = & \sum_{x\in X}\sum_{y\in Y}p_{x}f_{yx}\log\left(\frac{f_{yx}}{t_{y\nu (y)}f_{\nu (y)\mu (x)}^{\prime }}\right)\\ & + & \sum_{x\in X}\sum_{y\in Y}p_{x}f_{yx}\log\left(\frac{f_{\nu (y)\mu (x)}^{\prime }}{t_{\nu \left(y\right)\left(\nu^{\prime }\circ \nu \right)\left(y\right)}^{\prime }f_{\left(\nu^{\prime }\circ \nu \right)\left(y\right)\left(\mu^{\prime }\circ \mu \right)\left(x\right)}^{\prime \prime }}\right)\\ & = & \sum_{x\in X}\sum_{y\in Y}p_{x}f_{yx}\log\left(\frac{f_{yx}}{t_{y\nu (y)}f_{\nu (y)\mu (x)}^{\prime }}\right)\\ & + & \sum_{x^{\prime }\in X^{\prime }}\sum_{y^{\prime }\in Y^{\prime }}p_{x^{\prime }}^{\prime }f_{y^{\prime }x^{\prime }}^{\prime }\log\left(\frac{f_{y^{\prime }x^{\prime }}^{\prime }}{t_{y^{\prime }\nu^{\prime }\left(y^{\prime }\right)}^{\prime }f_{\nu^{\prime }\left(y^{\prime }\right)\mu^{\prime }\left(x^{\prime }\right)}^{\prime \prime }}\right)\;\;\left(\mathrm{by}\;\mathrm{item}\;\left(2\right)\;\mathrm{of}\;\mathrm{Lemma}\;2\right)\\ & = & \sum_{x\in X}p_{x}D\left(f^{x},{\left(t\circ f^{\prime }\circ \mu \right)}^{x}\right)+\sum_{x^{\prime }\in X^{\prime }}p_{x^{\prime }}^{\prime }D\left(f^{\prime x^{\prime }},{\left(t^{\prime }\circ f^{\prime \prime }\circ \mu^{\prime }\right)}^{x^{\prime }}\right)\\ & = & \mathrm{CE}(♣)+\mathrm{CE}(♠), \end{array}$$
as desired (the second-to-last equality follows by item (1) of Lemma 2). □

## 7. Relative Entropy in ${\mathrm{FinStat}}_{2}$

In this section, we introduce a 2-level extension of the relative entropy map RE introduced by Baez and Fritz, and show that it satisfies the natural 2-level analogues of functoriality, convex linearity, vanishing under optimal hypotheses, and lower semicontinuity.

**Definition** **13.***With every 2-morphism*
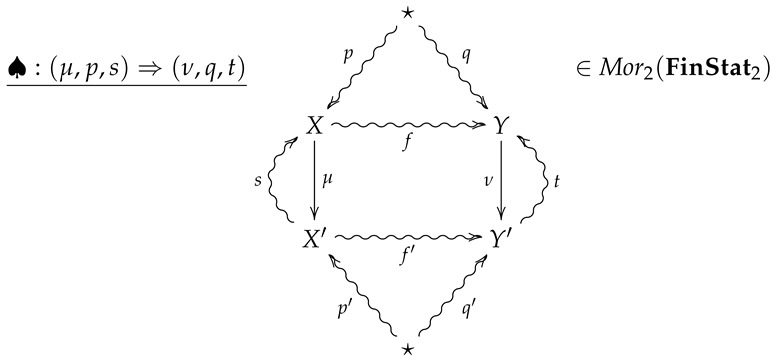

we associate the non-negative extended real number ${\mathrm{RE}}_{2}\left(♠\right)\in \left[0,\infty \right]$ given by
(8)$${\mathrm{RE}}_{2}\left(♠\right)=\mathrm{RE}\left(\mu ,p,s\right)+\mathrm{CE}\left(♠\right),$$
which we refer to as the ***2-relative entropy*** of ♠. We note that the quantity RE(μ,p,s) appearing on the RHS of (Equation 8) is the relative entropy associated with the morphism (μ,p,s) in FinStat, so that
RE(μ,p,s)=D(p,s∘μ∘p),
where D(−,−) is the standard relative entropy.

**Proposition** **2.**
*2-Relative entropy is convex linear, i.e., if (X,p) is a finite probability space and ${♠}^{x}$ is a collection of 2-morphisms in ${\mathrm{FinStat}}_{2}$ indexed by X, then*

$$RE_{2}\left(\underset{x\in X}{⨁}p_{x}{♠}^{x}\right)=\sum_{x\in X}p_{x}RE_{2}\left({♠}^{x}\right).$$



**Proof.** Suppose ⨁x∈Xpx♠x is summarized by the following diagram

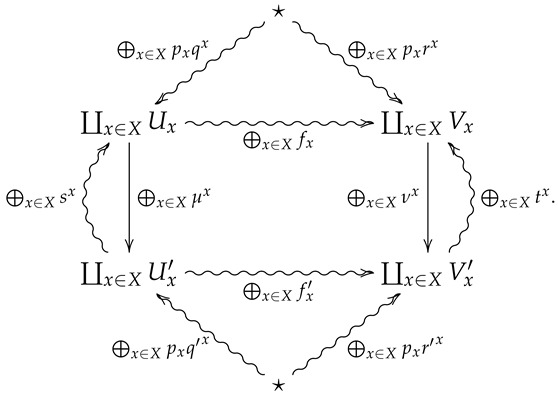

By Theorem 1, we know that the relative entropy RE is convex linear over 1-morphisms in ${\mathrm{FinStat}}_{2}$, and by Proposition 1, we know conditional relative entropy is convex linear over 2-morphisms in ${\mathrm{FinStat}}_{2}$, thus
(9)$$\begin{array}{c} \mathrm{RE}\left(\underset{x\in X}{⨁}p_{x}\left(\mu^{x},q^{x},s^{x}\right)\right)=\sum_{x\in X}p_{x}\mathrm{RE}\left(\mu^{x},q^{x},s^{x}\right)\;\&\;\mathrm{CE}\left(\underset{x\in X}{⨁}p_{x}{♠}^{x}\right)=\sum_{x\in X}p_{x}\mathrm{CE}\left({♠}^{x}\right). \end{array}$$
We then have
$$\begin{array}{ccc} {\mathrm{RE}}_{2}\left(\underset{x\in X}{⨁}p_{x}{♠}^{x}\right) & = & \mathrm{RE}\left(\underset{x\in X}{⨁}p_{x}\left(\mu^{x},q^{x},s^{x}\right)\right)+\mathrm{CE}\left(\underset{x\in X}{⨁}p_{x}{♠}^{x}\right)\\ & \overset{\left(9\right)}{=} & \sum_{x\in X}p_{x}\mathrm{RE}\left(\mu^{x},q^{x},s^{x}\right)+\sum_{x\in X}p_{x}\mathrm{CE}\left({♠}^{x}\right)\\ & = & \sum_{x\in X}p_{x}\left(\mathrm{RE}\left(\mu^{x},q^{x},s^{x}\right)+\mathrm{CE}\left({♠}^{x}\right)\right)\\ & = & \sum_{x\in X}p_{x}{\mathrm{RE}}_{2}\left({♠}^{x}\right), \end{array}$$
as desired. □

**Theorem** **3.**
*Relative entropy is functorial with respect to vertical composition, i.e., if ♣∘♠ is a vertical composition in ${\mathrm{FinStat}}_{2}$, then ${\mathrm{RE}}_{2}\left(♣\circ ♠\right)={\mathrm{RE}}_{2}\left(♣\right)+{\mathrm{RE}}_{2}\left(♠\right)$.*


**Proof.** Suppose ♠ and ♣ are such that the vertical composition ♣∘♠ is summarized by the following diagram.

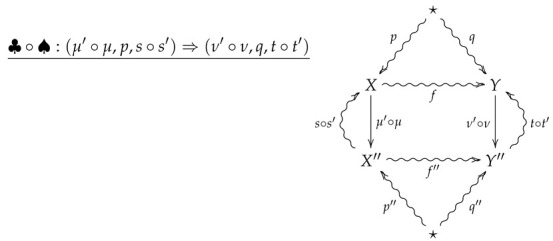

Then,
$$\begin{array}{ccc} {\mathrm{RE}}_{2}\left(♣\circ ♠\right) & = & \mathrm{RE}\left(\mu^{\prime }\circ \mu ,p,s\circ s^{\prime }\right)+\mathrm{CE}\left(♣\circ ♠\right)\\ & = & \mathrm{RE}\left(\mu ,p,s\right)+\mathrm{RE}\left(\mu^{\prime },p^{\prime },s^{\prime }\right)+\mathrm{CE}\left(♣\right)+\mathrm{CE}\left(♠\right)\\ & = & {\mathrm{RE}}_{2}\left(♣\right)+{\mathrm{RE}}_{2}\left(♠\right), \end{array}$$
where the second equality follows from Theorems 1 and 2. □

**Proposition** **3.**
*Let ♠:(μ,p,s)⇒(ν,q,t) be a 2-morphism in FinStat, and suppose s and t are optimal hypotheses for (μ|p) and (ν|q) as (defined in Definition 5). Then, ${\mathrm{RE}}_{2}\left(♠\right)=0$.*


**Proof.** Since *s* and *t* are optimal hypotheses, it follows that RE(μ,p,s)=CE(♠)=0, from which the proposition follows. □

**Proposition** **4.**
*The 2-relative entropy ${\mathrm{RE}}_{2}$ is lower semicontinuous.*


**Proof.** Since the 2-relative entropy ${\mathrm{RE}}_{2}$ is a linear combination of 1-level relative entropies, and 1-level relative entropies are lower semicontinuous by Theorem 1, it follows that ${\mathrm{RE}}_{2}$ is lower semicontinuous. □

## 8. Conclusions, Limitations and Future Research

In this work, we have constructed a 2-categorical extension ${\mathrm{RE}}_{2}$ of the relative entropy functor RE of Baez and Fritz [1], yielding a new measure of information which we view as a relative measure of information between noisy channels. Moreover, we show that our construction satisfies natural 2-level analogues of functoriality, convex linearity, vanishing under optimal hypotheses and lower semicontinuity. As the relative entropy functor of Baez and Fritz is uniquely characterized by such properties, it is only natural to question if our 2-level extension ${\mathrm{RE}}_{2}$ of RE is also uniquely characterized by the 2-level analogues of such properties. It would also be interesting to investigate alternative versions of 2-morphisms in FinStat where the 2-morphisms are less restrictive, such as where the base pyramid of a 2-morphism is not assumed to be commutative. While taking the 2-relative entropy associated with such morphisms would not be functorial, such less restrictive morphisms would provide more flexibility for potential applications. Finally, as there are many other categories of interest with respect to information theory [8,14,15], it would be interesting to investigate 2-level extensions of such categories as well.

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
