# Peer review of "On a 2-Relative Entropy"

_entropy, 2021, doi:10.3390/e24010074_

Round 1

Reviewer 1 Report

The paper needs minor revision. Please see the attached file.

Author Response

Dear Reviewer,

Thank you for your time and suggestions regarding our manuscript. We have added the references as suggested.

With kind regards,

James Fullwood

Reviewer 2 Report

Report on the manuscript "entropy-1517490" entitled "On a 2-Relative Entropy"

This manuscript constructs an extension of the Baez-Fritz relative entropy and shows that such a construction is functorial with respect to vertical morphisms, among other results. The construction is visualized as a relative measure of information between channels instead of relating it to probabilities. Conclusions about the present investigation are reported.

In general, I recommend a major revision of this manuscript that considers the following points:

1. The manuscript needs to be proofread by the author.

2. The title must be more explicit, informative, improved, and further extended for better clarity.

3. Words in the title are not usually in the keywords. In addition, the keywords are often written in alphabetical order.

4. The author must check the use of acronyms, abbreviations, and notations employed in the whole manuscript. 

5. In general, it is usual that section of the introduction presents (in the following order) the topic, motivations of the work, bibliographical review, objectives, the novelty of the manuscript, and description of its sections, with no formulas, which can be moved to a section of background on the topic. This organization must be considered in the revised manuscript.

6. I do not have each numerical result in detail. I recommend the author to check them.

7. The author must add an experimental study that allows their methodology to be applied and evaluated empirically. When applying the methodology, the author must provide details about the computational framework used. For example, software and packages employed, features of the computer utilized, and other computational aspects must be added. In addition, I recommend summarizing the methodology in an algorithm (and ideally in a flowchart) so that the readers can follow it easier.

8. In my opinion, the implications of the study are underdeveloped and must be improved and explained further in the final section.

9. The final conclusion needs to be improved. The author must add limitations of the study and more ideas for further research. Then, I suggest titling the final section as "Conclusions, limitations, and future research".

10. The bibliographical review must be improved significantly. Such it is, the review is totally insufficient and few informative for a scientific paper.

Author Response

Dear Reviewer,

Thank you for your time and thoughtful suggestions regrading our manuscript, as we believe our paper has significantly benefitted from your suggestions. In particular, we have made the following changes according to your suggestions:

  1. We have identified and fixed typos.
  2. We have made the title more explicit and informative.
  3. We have alphabetized keywords.
  4. We have made sure our notations and use of acronyms are consistent.
  5. In the second to last paragraph of the introduction we have included a more detailed outline of the paper.
  6. As we don't have any numerical results in the paper, we were unable to address the issue of giving numerical results in more detail.
  7. We have added Remark 23 and Example 24 in Section 4 to address how our construction of 2-morphisms in FinStat_2 may be realized empirically.
  8. We have extended the final section to address implications of the study. 
  9. We have updated the name of the final section which better reflects it updated contents.
  10. We have added 66.7% more references.

With kinds regards,

James Fullwood

Reviewer 3 Report

Please see my report.

Author Response

Dear Reviewer,

Thank you for your comments regarding my manuscript. As you suspect, category theory is not needed to express Baez and Fritz' characterization of relative entropy or the results in my paper. This is mainly due to the fact that we do not even use category theory beyond calling certain sets "objects" and certain maps between such sets "morphisms" (and also maps between maps "2-morphisms" in my manuscript). As such, a natural question is then why are we using category theory at all? Well, there are two main reasons.

  1. It is often not clear how to generalize many aspects of classical information theory to the quantum realm, and category theory provides an indispensable tool for doing so. In particular, there is a certain functor (or map) which embeds classical information theory into quantum information theory, and if we phrase our results in the language of categories, then it will be straightforward to translate out results into the quantum regime. For a specific example, my collaborator Arthur Parzygnat has carried out this program with Baez and Fritz' characterization of relative entropy to obtain quantum analogues of their results (see arXiv:2105.04059 ). 
  2. Category theory is being used more and more prevalently in applied mathematics, so much so that there is even a wIkipedia page on applied category theory, and there is also an annual conference at MIT on the subject. In my view this emerging field of applied category theory is a consequence of the fact that category theory provides unifying language in mathematics, and once one is comfortable with such a language it is difficult not to think in such terms.

So while it is certainly possible to phrase our results without using the language of categories, we do so precisely for the benefit of generalizing to both the quantum realm and also to higher categorical structures. We also state at the beginning of Section 2 that "Though we use the language of categories, knowing that a category consists of a class of composable arrows between a class of objects is sufficient for the comprehension of all categorical notions in this work."

To address your specific points: 

  1. I fixed the typo "In such as case" to "In such a case".
  2. The map s is a Markov kernel from Y to X . The condition f\circ s is the identity on Y ensures that s_{xy}=0 if f(x) is not equal to y. The map s is also sometimes called a "disintegration"  by probability theorists. The specific interpretation of s in our setting is addressed in Remark 11.
  3. f' is indeed a map from X' to Y'.
  4. CE can also be viewed as a relative measure of information between channels, but without adding the term RE(\mu,p,s), it does not use the information of s at all, and as such is not incorporating all the data of a 2-morphism. I call CE the "conditional relative entropy" as its definition has a very similar form to conditional entropy (see equation 25 and Remark 26). You can also think of CE as the convex sum on the RHS of equation (1), which is the chain rule for relative entropy.   

Round 2

Reviewer 2 Report

The authors did address my main concerns, that is,

1. In general, it is usual that section of the introduction presents (in the following order) the topic, motivations of the work, bibliographical review, objectives, the novelty of the manuscript, and description of its sections, with no formulas, which can be moved to a section of background on the topic. This organization must be considered in the revised manuscript.

2. The author must add an experimental study that allows their methodology to be applied and evaluated empirically. When applying the methodology, the author must provide details about the computational framework used. For example, software and packages employed, features of the computer utilized, and other computational aspects must be added. In addition, I recommend summarizing the methodology in an algorithm (and ideally in a flowchart) so that the readers can follow it easier.

3. In my opinion, the implications of the study continue to be underdeveloped and must be improved and explained further in the final section.

4. The final conclusion needs to be improved.

5. The bibliographical review must be improved. Such it is, the review is totally insufficient and few informative for a scientific paper.

Author Response

Dear Reviewer,

There seems to be a misunderstanding regarding my manuscript. In particular, my manuscript is a mathematics paper, and not an empirical scientific study. Moreover, it is my understanding that the journal Entropy is multi-disciplinary, and as such, publishes both scientific papers as well as mathematics papers. For example, one may consult the mathematics papers

  1. John C. Baez, Tobias Fritz, and Tom Leinster, A characterization of entropy in terms of information loss, Entropy 13 (2011), no. 11, 1945–1957
  2. Imre Csiszar, Axiomatic Characterizations of Information Measures, Entropy 10 (2008), no. 3, 261-273
  3. James Fullwood and Arthur J. Parzygnat, The Information Lass of a Stochastic Map, Entropy 23 (2021), no. 8, Paper no. 1021
  4. Pierre Baudot and Daniel Bannequin, The Homological Nature of Entropy,                   Entropy 17 (2015), no.5, 3253-3318,

and find that they all do not adhere to many of the conditions that you are requiring my manuscript to meet, such as

  • no formulas in the introduction
  • empirical studies which allow the methodology to be applied and evaluated empirically
  • algorithms summarizing methodology
  • software packages employed

Further still, if one reads the actual content of my manuscript, they will find that it wouldn't even make sense to have algorithms summarizing methodology, or software packages employed. In particular, the formula given in the definition for 2-relative entropy is a convex combination of relative entropies, and as such, doesn't require any new methods for its computation. 

In any case, my manuscript is a mathematics paper which introduces a new measure of information which may be viewed as a relative measure of information between discrete memoryless channels. To make more contact with the real world and empirical studies, I have added Remark 23 which gives a general operational interpretation of my construction, and I have also added Example 24 to include a real-life scenario which realizes my construction. I make no claim that my paper is a scientific study in any way, and as such, it should not be treated as one.

With kind regards,

James Fullwood                                                                

Round 3

Reviewer 2 Report

The authors did not respond to my concerns. As long as they are not addressed, my decision is rejection.

Author Response

N/A